# Evaluation of the New Beckmann Coulter Analyzer DxH 900 Compared to Sysmex XN20: Analytical Performance and Flagging Efficiency

**DOI:** 10.3390/diagnostics11101756

**Published:** 2021-09-24

**Authors:** Maite Serrando Querol, Javier Nieto-Moragas, Anna Marull Arnall, Meritxell Deulofeu Figueras, Orlando Jiménez-Romero

**Affiliations:** 1Hematology Laboratory, Hospital Universitari de Girona Doctor Josep Trueta, 17190 Girona, Catalonia, Spain; jnieto.germanstrias@gencat.cat (J.N.-M.); amarull.girona.ics@gencat.cat (A.M.A.); mdeulofeu.girona.ics@gencat.cat (M.D.F.); ojiminez.bnm.ics@gencat.cat (O.J.-R.); 2Research Group of Clinical Anatomy, Embryology and Neuroscience (NEOMA), Department of Medical Science, Faculty of Medicine, Universitat de Girona, 17003 Girona, Catalonia, Spain

**Keywords:** cell blood count (CBC), leukocyte differential, Beckman Coulter DxH 900, Sysmex XN20

## Abstract

Efficiency and accuracy in automated hematology analyzers are very important for clinical laboratories. The purpose was to evaluate the flags and results reported by the newest Beckman Coulter analyzer DxH 900 compared to the Sysmex XN20 system. Samples were analyzed on the XN20 (Sysmex, Kobe, Japan) and on the Beckman Coulter DxH 900 (Beckman Coulter, Miami, Florida, USA). Slide reviews were performed microscopically. Morphologic criteria were used to identify abnormal cells as recommended by International Consensus Group for Hematology (ICSH): blasts, immature granulocytes (IG%), abnormal lymphocytes (ALs) and plasma cells. Results: there was a strong correlation between the analyzers in almost all clinical parameters tested. Both DxH 900 and XN20 showed an excellent degree of association for the leukocyte differential compared to the reference method (manual microscopy). When it comes to IG%, XN20 showed a positive bias for higher results. Related to platelets, there are no differences between the two methods for PLT count. For mean platelet volume (MPV), DxH 900 provided 100% results of the samples analyzed while XN20 while in the XN20 analyzer, 16% of the results were missing. From our results we came to the conclusion that both analyzers, DxH 900 and XN20 were clinically accurate and efficient. Abnormal Lymphocyte detection highlighted the differences between the two technologies as only minimal agreement was obtained. DxH 900 demonstrated higher sensitivity in detecting IG with good correlation with microscopic review. The DxH 900 for platelet clumps identification provides an excellent flag (PLT Clumps) with the highest sensitivity observed in our evaluation.

## 1. Introduction

For modern clinical laboratories with a heavy workload, efficiency and high-quality results are the most valuable characteristics to consider in automated hematology analyzers [1]. These systems provide high speed and accuracy to laboratories thanks to their developed software. Modern analyzers contain detection systems which can identify abnormal cells making the diagnosis more accurate [1,2,3]. However, in some cases peripheral blood smear observation is required. This process is time-consuming and requires a high level of expertise which is not present in all laboratories. The latest generation of analyzers provides additional information such as research use only parameters (RUO) and cell population data (CPD) which increases the detection of pathological cells, and therefore, improves the diagnosis of hematological diseases.

The DxH 900 is the newest hematology system available from Beckman Coulter. It is able to perform a cell blood count (CBC) with sophisticated technology improving the accuracy of platelets identification (PLTs), white blood cells count and differential (WBCs), red blood cells (RBCs), immature granulocytes (IGs) and nucleated red blood cells (NRBCs). The CBC module is based on the enhanced Coulter principle using digital impedance with triplicate counting, voting and sweep flow in order to prevent the recirculation of cells behind the aperture. The WBC differential, NBRC count and IG identification is performed with VCS^n^ technology (V: volume; C: conductivity; S: scattergram), which applies several physical measurements to each individual cell. These measurements include impedance to obtain cell volume, conductivity in radiofrequency current to analyze internal composition of the cell and nucleus-to-cytoplasm ratio and five light-scatter measurements to acquire information about cellular granularity. Cells are analyzed in near-native state, an environment as close to the human body as possible and without any stain. The DxH 900 flagging system for the presence of abnormal cells is based on multiple algorithms, including snake analysis and template matching. The resultant scattergrams are compared to a normal pattern of distribution obtaining a matching score with three levels of confidence: low, medium or high. The DxH 900 technology provides detailed cellular analysis with 70 cellular morphometric data (CMD), which can be used in decision rules to enhance the pathology-screening capabilities of the analyzer. Data fusion increases confidence in results and improves flagging, eliminating the need for reruns. 

The Sysmex XN20 is a multiparameter hematology counter which uses a combination of impedance and flow cytometry for the CBC and the differentiation of blood cells [4,5]. The XN20 leukocyte differential is based on the fluorescence intensity of the cells and cell complexity after chemical modification of the cell membrane. The XN20 analyzer has an adaptive flagging algorithm based on shape recognition (SAFLAS: Sysmex adaptive flagging algorithm) which builds a linear discriminate by combining the results of 6 different parameters of each cell population cluster and angle of cell cluster (total cell population; population size, width cluster population; cluster population position; shape of cell cluster and angle of cell cluster. This analysis enables the XN20 to flag samples. 

The aim of this study was first to evaluate and compare the analytical performance of the two systems by determining CBC and WBCs differentiation. Secondly, to evaluate the flagging capabilities and efficiency in the detection of abnormal cells (atypical lymphocytes, abnormal lymphocytes, IG% and blasts) when compared to the reference method (manual microscopy) and to each other. 

## 2. Materials and Methods

In this study 1000 K3-EDTA-anticoagulated blood samples (Becton Dickinson, Plymouth, UK) were included and analyzed in the hematology laboratory department of Parc Martí i Julià Hospital (Salt, Spain). The testing of all samples for this study was completed within 2–4 h after the phlebotomy. In this study, 53% of samples were from women and 47% were from men. The age distribution of this group of patients was: 12–20 years-old (8.5%); 21–40 years-old (14.3%); 41–60 years-old (25.6%); 61–80 years-old (35.9%) and finally ≥81 years-old (17.5%). Distribution of the patients by origin was: primary care 57%, admitted patients (including oncology, hematology, internal medicine and surgery) 38% and non-admitted oncology-hematology patients 5%. 

The analyzers were calibrated using stabilized calibration material from Beckman Coulter and Sysmex for the respective systems. The results from samples included in the study were integrated using appropriate and compatible software for each analyzer: DxH 900 SMS II 1.0 (Beckman Coulter, Miami, FL, USA) for the Beckman Coulter analyzer and Information Process Unit (IPU) (Sysmex, Kobe, Japan) for the Sysmex XN20. 

We used R studio open software for statistics. Passing-Bablok regression and Bland-Altman analysis were performed to compare both analyzers (results from the two analyzers are plotted against the median of the two techniques). Spearman’s test was used to obtain the correlation coefficient. For the basophile count we performed a Chi test. 

### 2.1. Inclusion/Exclusion Criteria

Inclusion criteria were: the inclusion of different ages according to the groups of the study; the inclusion of different genders, races and ethnicities; the inclusion of all disease states. Exclusion criteria: samples that were visibly clotted and samples with insufficient volume. Samples not meeting handling instructions and samples run outside of instrument quality control were also excluded. We excluded 13 samples due to the lack of CBC or manual count results. 

Slide reviews were performed in all the samples as reference method. Blood smears were prepared for each sample using the Sysmex SP-1000i equipment (Sysmex, Kobe, Japan) and stained using the May-Grünwald method. Differential count was performed using the CellaVision DM96 digital microscope (CellaVision, Lund, Sweden) (200 cells/smear, two smears/patient) and reviewed by two different observers in accordance with Clinical and Laboratory Standards Institute document CLSI H20-A2 [6]. The classification and identification of the cells was performed by physicians with significant expertise in cytology. 

Morphologic criteria were used to identify abnormal cells recommended by International consensus group for hematology (International society for laboratory hematology, ISLH) [5]: blast ≥1%, immature granulocytes (IG%: myelocyte/promyelocyte >1%, metamyelocytes >2%) and abnormal lymphocytes ≥5% and plasma cells ≥1%. In order to be more accurate in terms of flag performance, we included retrospectively different cut-off levels for blast and abnormal lymphocytes. We defined thrombocytopenia as platelet count (PLT) below <100,000/μL [7]. In order to be more focused on the clinical condition of the patients we added a second cut-off for samples below <60,000/µL [5,7]. 

### 2.2. Analysis of the Flagging Performance

We analyzed the flagging performance in the DxH 900 and the XN20 in the presence of abnormal cells. We included common analyzer flags such as blast, immature granulocytes (IG%) and platelet clumps. For the detection of abnormal lymphocytes each analyzer has specific flags. The variant lymph high sensitivity flag was used for the DxH 900 whereas blast/abn lymph and atypical lymph flags were used for the XN20. If the blast/abn Lymph is present, the XN20 performs subsequent analysis in the white precursor cell channel (WPC) that allows the analyzer discrimination between blast and abnormal cells detection. In this study these two XN20 flags were assessed together to ensure homogeneity with DxH 900 that only has one flag for the abnormal lymphocyte detection. We did not included flags from de WPC in the XN20 performance. 

### 2.3. Analytical Performance and Statistical Procedure

Between-day (three levels of quality control samples) and within-day (three levels of different patient samples) and precision were evaluated for WBC, RBC, hemoglobin (Hb) and PLT for different values. Passing Bablok regression was performed to evaluate the agreement between analyzers using patient samples. The Cusum test and Spearman rank correlation coefficient were applied in order to check the linearity and correlation between the DxH 900 and the XN20. For slope and intercept, we calculated 95% confidence intervals for each parameter using the bootstrap method. Expected bias was estimated at selected threshold values and Bland Altman analysis was met to detect tendencies over the range of values [8]. 

## 3. Results

### 3.1. Precision Essay

Precision essays were performed according to CLSI-EP9 for Clinical and Laboratory studies [6]. Between-day analysis compared three levels of quality control (QC to the manufacturers and state of the art precision limits). No differences were observed between values obtained from either analyzer (*p* > 0.05) (Table 1).

In the within-day analysis, we observed coefficients of variation (CV%) lower than 5% for each parameter tested (Table 2). The CV% for this analysis was lower than 5% for all the parameters tested except for the low platelet measurement with the XN20 (CV = 9.99%), showing higher differences than ought to be observed. However, relevant variations related to platelet count were observed with medium concentration, showing significant differences between the two analyzers. The XN20 impedance method reported slightly higher results than the DxH 900.

### 3.2. Agreement between Analyzers

Correlation and regression analysis were carried out on measurements from each analyzer against the median. The correlation among the analyzers was good according to the results of the Spearman Rho coefficients displayed in Table 3. However, the DxH 900 was significantly different from the XN20 due to a proportional difference between the two methods as the confidence interval for slope did not contain the value 1. Moreover, both methods differed by a constant amount as the confidence interval for intercept did not contain the value 0. Therefore, the methods were not entirely interchangeable. 

Despite slight differences observed, almost all clinical results showed a strong correlation, except WBC and MPV in normal and pathological range. The Bland Altman plot for WBC of both analyzers showed that the XN20 count tended to be larger as the average increased (Figure 1). 

PLT and RBC indices were measured by impedance methodology in both analyzers. Variability was consistent over the range of values for all the parameters tested. No systemic errors were observed for hemoglobin, MCV, MCH and PLT. As Bland Altman showed in Figure 2, for MPV the XN20 counted 2 fL higher than the DxH 900 for both high and low results. The DxH 900 provided results of MVP in 100% of the samples analyzed while the XN20 had a 16% result of that parameter left.

### 3.3. Analyzers and Reference Method Comparison for Leukocyte Differential

Leukocyte differential was performed by both analyzers on 987 samples with the results being compared to the results obtained using the reference method. We included samples with suspect messages from the analysis related to abnormal cell detection such as blast, IG% (immature granulocytes) or var lymph (abnormal lymphocytes) on the DxH 900 and blast/abn lymph (blast and abnormal lymphocyte) or atypical lymph (reactive lymphocytes) on the XN20. Passing Bablock correlation between the DxH 900 and reference method (optic microscopy) can be seen in Table 4. Passing Bablok correlation between the XN20 and reference method can be seen in Table 5. 

We performed neutrophil, lymphocyte and monocyte correlation between the DxH 900 and the XN20 against those obtained with the reference method. The degree of association for neutrophils, lymphocytes and eosinophils showed a significant Spearman Rho value in both the DxH 900 (from Table 4, 0.940, 0.918 and 0.834, respectively) and the XN20 (from Table 5, 0.943, 0.927 and 0.833, respectively). For monocytes, correlation showed a discrepancy with the reference method (Rho Spearman value of 0.700 for the DxH 900 and 0.693 for the XN20). Basophile count distribution did not allow building a regression so we performed the Chi test with good kappa test value for both analyzers (K = 0.234 for the DxH 900 basophile count, K = 0.277 for the XN20 basophile count). 

### 3.4. Immature Granulocytes (IG%) Agreement between Analyzers and Reference Method

We calculated sensitivity and specificity of IG% by comparing the results from both analyzers to the reference method (optic microscopy). We included samples with IG > 2% and IG > 5%, including infectious and hematologic malignant diseases. For the first condition tested (IG > 2%) sensitivity in the DxH 900 was 85.4% and in the XN20 78%. In those samples with IG > 5%, sensitivity in the DxH 900 was 91.7% and in the XN20 83.3%. Some true positives samples may be included due to toxic granulations or hyposegmentation of neutrophils. 

### 3.5. Clinical Sensitivities of Abnormal Cell Flags

Total flag distribution in our study was 19.7% in the DxH 900 and 24.1% in the XN20. In the DxH 900 the most frequent flag observed was immature granulocytes (9.2%) followed by abnormal lymphocyte (7%) and blast (5.7%). In the XN20 the most common flag detected was blast/abn Lymph (15.2%) followed by immature granulocytes (8.7%).

#### 3.5.1. Blast Detection

Morphological blast flags for the DxH 900 and blast/abn lymph for the XN20 were compared using the microscopic criteria defined in the study. Table 6 shows sensitivity and specificity results for both analyzers considering three different cut-offs. In terms of sensitivity and specificity, with the first condition studied (>1%), the XN20 had better results than those observed with the DxH 900. Second condition studied (>5%) showed similar results for both analyzers by detecting 7/8 samples with blasts with the DxH 900 and 8/8 with the XN20. The last condition studied (blast >20%), which is the World health organization (WHO) blast cut-off for acute leukemia (AL), maintained good results for the DxH 900 and the XN20. As we can observe in Table 6, the XN20 showed for the three conditions tested, a higher number of false positive cases than the DxH 900. 

#### 3.5.2. Abnormal Lymphocyte Detection

The DxH 900 flag evaluated was variant high sensitivity lymph. For the XN20 we evaluated two flags for abnormal lymphocytes: atypical lymphocytes (related to reactive lymphocytes) and blast/abnormal lymphocytes. From the microscopic count, in this study 44% of the samples analyzed showed abnormal lymphocytes, either for reactive lymphocytes, non-Hodgkin lymphoma or B-cell chronic diseases. 

In terms of abnormal lymphocyte detection (Table 7) sensitivity for the DxH 900 flag was 11.9%, specificity 97.3% and balanced accuracy of 54.6%. On the other hand, for the XN20 flag, sensitivity was 24%, specificity 90% and balanced accuracy 57%. Considering that with this approach we might be mixing two different types of cells, reactive and abnormal lymphocytes, we checked on the results considering the following morphological conditions: reactive lymphocytes >5% or abnormal lymphocytes >5% (either abnormal lymphocytes or plasma cells). Taking as first condition >5% of abnormal cells in peripheral blood smear, we considered three different situations: lymphocytosis >3.5 × 10^9^/L, lymphocytosis >5 × 10^9^/L and relative lymphocytosis >50%. As Table 7 shows, both analyzers provided good sensitivity (DxH 900 = 93.8% and Sysmex XN = 87.5%) when we observed relative lymphocytosis (>50%). Moreover, we observed good balance accuracy for the DxH 900 (90.1%) flag when absolute lymphocytosis was present. In Table 7, likewise we observed with blast detection performance, the XN20 showed a higher number of false positive results than the DxH 900 for all the conditions tested. 

Detection of reactive lymphocytes was analyzed using the same conditions as those for abnormal lymphocyte performance (reactive lymphocytes >5% and lymphocytosis >3.5 × 10^9^/L, reactive lymphocytes >5% and lymphocytosis >5 × 10^9^/L and reactive lymphocytes >5% and lymphocytosis >50%). We observed a sensitivity of 9.9% and 10.6% for the DxH 900 using the first and second conditions, respectively. Sensitivity values for the XN20 were 1.4% (first condition tested) and 1.5% (second condition tested). According to the results displayed in Table 8, there were more false positives cases provided by the DxH 900 than by the XN20. 

#### 3.5.3. Platelet Clumps Analysis

As we can observe in Table 9, two different clinical conditions were considered for thrombocytopenia combined with observation of platelet clumps in the peripheral blood smear [7]: first, platelets <100,000/µL and second, platelets <60,000/µL. The DxH 900 sensitivity for the first condition was 85% achieving 100% with the second condition studied. Specificity in both conditions was 98.5% for the first condition and 97.7% for the second. The XN20 platelet clumps performance showed 30% and 50% of sensitivity, respectively, for the two conditions studied. In addition, specificity in both conditions was 99% for the first condition and 98.8% for the second.

## 4. Discussion

Most current automated hematological analyzers provide blast detection as well as NRBC, immature granulocytes and abnormal lymphocyte detection, which are fundamentally related to non-neoplastic or neoplastic hematological diseases [9,10,11]. The newest analyzer DxH 900, has improved the results obtained with its predecessor (DxH 800) for leukocyte differential [3,12]. Sysmex XN system technology, with impedance and fluorescent flow cytometry provides three signals for every cell and has extra channels for the rerun tests needed [1,9,13]. In order to be more specific, automated systems have different types of flags that are recognized as a warning, indicating that peripheral blood smear review is needed. This process requires time and a high level of cytology expertise, slowing down the diagnosis considerably [10,14]. Moreover, some flags do not show enough sensitivity and therefore should be considered as inconsistent or intended for orientation only. For this reason analytical and flag performance of these instruments should be studied. 

For cell blood count parameters, the between day and within run coefficients of variation (CV%) on both analyzers were lower than 5%. The platelet CV% though was higher than expected in the between day imprecision analysis for the XN20. Despite this observation, there is no clinical difference between the two of them. Similar results have been reported in other published studies [1,3,10]. It should also be noted that platelet count CV% in the within run imprecision analysis were excellent for both analyzers. According to B.D. Hedley et al. [2] The DxH system provides accurate platelet results with an excellent agreement with the flow reference method (CD41+CD61) and better precision compared to other instruments [7,8,15]. 

Correlation studies between the two analyzers showed good results with bias towards for CBC parameters according to the spearman Rho coefficient for RBC, hemoglobin, Hematocrit, MCV, RDW and PLT [9,10,16]. There was not a clinical transcendence in the differences between the two instruments. Thus, they can be used indistinctly as almost all results demonstrated a strong correlation [13]. For WBC measurements, both analyzers show similar results except in high count of cells (>10 × 10^3^/µL) where the XN20 tends to count more cells than the DxH 900. For platelets, the XN20 has a negative bias observed in all samples analyzed. Moreover, in order to check on abnormal platelet count or abnormal platelet distribution, the XN20 needs PLT-F extra channel. This is time consuming and needs specific reagents. With regards to MPV, difference between 2 analyzers is expectable as there is no reference method for MPV and therefore MPV results between 2 instruments cannot be standardized. Absence of MPV standardization methods makes the comparative analysis more difficult [8,15]. From the Bland Altman display we concluded that the XN20 counted 2 fL higher than the DxH 900 for both high and low results, meaning that we will always observe higher MPV in the XN20. In addition, the XN20 system did not provide MVP results for the totality of the samples analyzed as the DxH 900 system did. In this respect, this lack of results could be solved using the platelet fluorescent channel (PLT-F). 

A strong agreement was observed between the DxH 900 and the XN20 when it comes to leukocyte differential in accordance with other studies [1,3]. However, other studies conducted reveal better correlation coefficients using the Sysmex XN 3000 and the Unicel DxH 800 for basophile and monocytes [1,4,17]. Our results are in strong agreement with those reported by Meintker et al. while comparing the DxH 800 and the Sysmex XE [18]. Compared to the reference method, the DxH900 and the XN20 had an excellent degree of association for neutrophils and lymphocytes. For monocytes, the coefficient value was 0.720 in the DxH 900 and 0.693 in the XN20; these results were lower than initially foreseen, but could be justified due to the heterogeneity of monocyte morphology and the difficulty in their correct identification from large lymphocytes [19,20]. In this respect, Passing Bablock graphs showed an excellent degree of correlation between the two analyzers; results comparing the DxH 900 and the XN20 leukocyte differential values with the reference method were excellent, according to similar studies performed with Sysmex Series and DxH system [2,4,13]. 

One major point of concern in clinical laboratories with automated analyzers is flagging and effectively detecting those samples which require manual microscope review in order to improve sensitivity and specificity for abnormal cell identification. From our comparison we concluded that the XN20 showed a clear positive trend providing higher results for immature granulocytes (IG%) count. Better agreement for the DxH 900 was observed in all the cases, showing higher sensitivity than the XN20 detecting IG% with good correlation with manual count. The XN20 overestimated IG% particularly in those samples with IGs% > 5%. This observation disagrees with previous studies that showed a better sensitivity for Sysmex XN compared to the DxH 800 for IG% [1,12,21]. Technical modifications in the automated intelligence module by incorporating RUO early granulocytic cell (EGCs) parameters and software improvements in the DxH 900 have been crucial to increasing its sensitivity. 

Acute leukemia is one of the worst hematological conditions and therefore has to be detected quickly and correctly. It must be considered that many disorders could present blasts in peripheral blood, so there is an overlap between different kinds of hematological and non-hematological disorders [17,22]. In the laboratory, we should identify if the blast flag could lead to leukemia; overlapping interferences should be considered due to different disorders [17,18]. Moreover, morphology of the blast cells is often heterogeneous meaning that classification may be difficult without extensive expertise. Both analyzers showed good results in terms of sensitivity and specificity in blast detection. The XN20 obtained better results with the first condition tested even though this condition should be clinically considered as extremely strict. Using the World health organization (WHO) blast cut-off for acute leukemia (AL) defined as blast >20% in peripheral blood and/or bone marrow, we maintained good results for both analyzers despite the higher false positives detected by the XN20. Laboratory decision rules are especially important as they will help avoid missing serious pathologies, such as acute leukemia. When combining blast flag presence and conditions such as rules anemia, thrombocytopenia and/or leucocytosis or pancytopenia we obtained 100% of sensitivity and specificity. There was a notable difference in the false positive results though; the XN20 showed a significant number of false positive cases. These results can be justified according to the fact that the XN20 performs a retest in WPC channel in those samples with blast/abn lypmh flag. This retest provides additional measurements leading to the presence of two differentiated flags: blast or abn lymph. We had studied though a limited population of 18 samples which did not allow reaching a significant difference between the analyzers and reference method. Further investigations with a higher number of positive cases should be carried out. 

Morphological abnormalities in lymphocytes are difficult to identify with the microscope due to the high variety of morphological features which may be present. Most of the current analyzers have a single flag (abnormal or variant lymph) to detect the non-neoplastic condition as well as the neoplastic disorders. Therefore it is a challenge to improve on microscopic identification and description of the lymphocytes with a single analyzer flag. In abnormal lymphocyte detection, the slight agreements between the two analyzers reflected the different technologies used. Better flagging sensitivity for ALs was observed using the DxH 900 than with the XN20. Depoorter et al. reported better flagging sensitivity for the Unicel DxH 800 in the detection of ALs [14]. Results for the XN20 flag were not as good as those observed with the DxH 900, potentially due to the need for the retest in the WPC channel. The XN20 performs an extra analysis in the WPC channel in all the samples flagged with blast/abn Lymph. Thus, without this additional information required for the XN20 flagging algorithm we obtained a high number of false positive cases. When it comes to reactive lymphocyte detection, considering the same three conditions used beforehand, the DxH 900 shows better results in all the conditions tested than the XN20. We observed though more false positive cases in the DxH 900 performance than in the XN20 due to the better accuracy of the atypical lymphocyte flag. From a morphological point of view, abnormal lymphocyte flag in the DxH 900 leads to a general suspicion of lymphoid neoplasms or benignant diseases with reactive changes in lymphocytes.

Platelet disorders are classified as quantitative (lower or higher count) or qualitative (inherited or acquired) disorders [23]. It should be remembered that aggregates or dilution in samples can interfere with the quantitative results. We therefore have different clinical disorders with thrombocytopenia dependent on the patient condition [8,15]. Results provided by the XN20 showed that, when it comes to thrombocytopenia with platelet clumps flag, the best option was to perform a smear review. The DxH 900 sensitivity was 100% (platelets <60,000/µL) so platelet clumps on the DxH 900 was an excellent flag with the highest sensitivity observed in our evaluation.

According to our results, the DxH 900 and the Sysmex XN20 are accurate and efficient. Both can be considered precise systems and can operate effectively in laboratories with a high workload. The DxH 900 shows better results in detecting IGs, ALs and platelet clumps. In blast detection both analyzers show similar results, which allow high-volume laboratories to detect serious diseases such as leukemia. In this study we reviewed all the smears microscopically so our results might be considered a good comparison to the reference method in all the pathological conditions tested. 

## Figures and Tables

**Figure 1 diagnostics-11-01756-f001:**
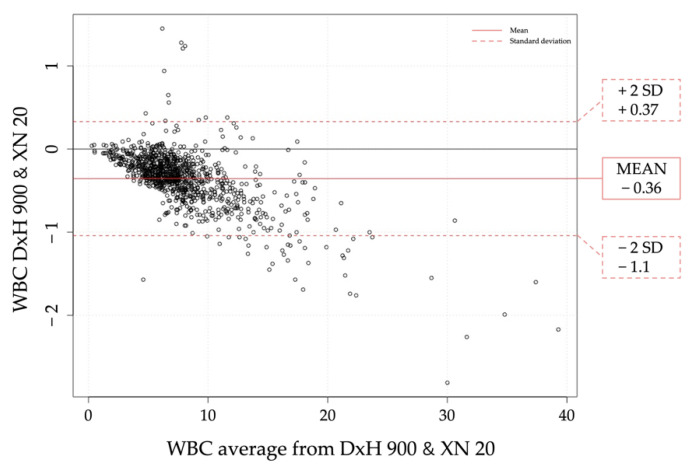
Bland Altman plot comparing WBC in both analyzers. Results from the two analyzers are plotted against the median of the two techniques. Solid point line is related to the mean of the results; hollow point line is related to positive and negative standard deviation.

**Figure 2 diagnostics-11-01756-f002:**
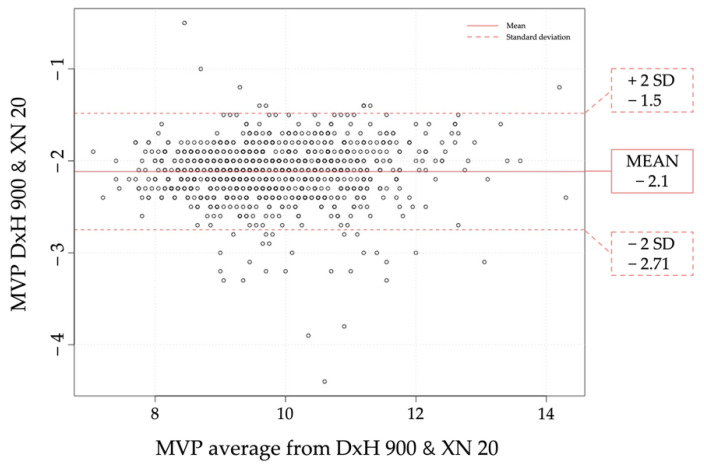
Bland Altman plot comparing MVP in both analyzers. Results from the two analyzers plotted against the median of the two techniques. Solid point line is related to the mean of the results; hollow point line is related to positive and negative standard deviation.

**Table 1 diagnostics-11-01756-t001:** Coefficients of variation (CV%) for between-day imprecision analysis during 25 days.

Within-Run Imprecision		Low Level	Medium Level	High Level
WBC (10^9^/L)	DxH 900	2.55 (3.4)	1.96 (8.03)	1.22 (20.7)
	XN20	1.8 (2.87)	1.8 (6.95)	0.8 (16.29)
RBC (10^12^/L)	DxH 900	1.83 (1.79)	1.59 (3.93)	2.09 (5.23)
	XN20	1.1 (2.36)	0.7 (4.41)	0.7 (5.35)
Hb (g/dL)	DxH 900	1.28 (4.8)	1.04 (11.8)	1.23 (15.4)
	XN20	1.1 (6.2)	0.8 (12.4)	0.5 (16.3)
Hct (%)	DxH 900	1.4 (15.0)	0.88 (35.9)	1.39 (47.7)
	XN 20	1.7 (18.6)	1.4 (37.5)	1.3 (48.8)
MCV (fL)	DxH 900	1.31 (83.5)	1.12 (91.2)	1.21 (91.3)
	XN20	1.4 (81.3)	1.2 (89.6)	1.31 (92.1
MCH (pg)	DxH 900	0.92 (26.8)	1.11 (29.4)	0.85 (30.1)
	XN20	0.8 (26.2)	0.98 (28.4)	1.05 (30.8)
PLT (10^9^/L)	DxH 900	3.02 (68)	1.99 (208)	1.65 (389)
	XN20	4.1 (82)	4.8 (241)	2.8 (548)

WBC: White blood cells; RBC: Red blood cells; Hb: Haemoglobin; Hct: Hematocrit; PLT: Platelets; MVC: mean corpuscular volume; MCH: mean corpuscular hemoglobin. In the table CV% and mean for all parameters tested (Normal distribution of all CV% results included).

**Table 2 diagnostics-11-01756-t002:** Coefficients of variation (CV%) for within-day imprecision analysis for 10 repetitions in a day.

Between-Day Imprecision		Low Level	Medium Level	High Level
WBC (10^9^/L)	DxH 900	0.99 (2.98)	1.79 (7.38)	0.64 (17.34)
	XN20	1.61 (3.02)	1.93 (7.5)	1.33 (17.84)
RBC (10^12^/L)	DxH 900	0.42 (3.17)	0.27 (4.30)	0.78 (4.60)
	XN20	0.82 (3.27)	0.42 (4.39)	0.54 (4.69)
Hb (g/dL)	DxH 900	0.39 (9.57)	0.33 (10.6)	0.27 (12.23)
	XN20	0.51 (9.43)	0.70 (10.6)	0.43 (12.1)
Hct (%)	DxH 900	0.5 (29.0)	0.86 (34.4)	0.59 (37)
	XN 20	0.83 (29.5)	0.41 (34.9)	0.38 (36.3)
MCV (fL)	DxH 900	0.40 (74.7)	0.48 (86.5)	0.34 (91.5)
	XN20	0.19 (74.4)	0.18 (82.8)	0.15 (91.3)
MCH (pg)	DxH 900	0.95 (23.1)	0.28 (28.4)	0.32 (30.1)
	XN20	0.54 (22.6)	0.44 (27.6)	0.64 (28.8)
PLT (10^9^/L)	DxH 900	2.39 (54)	1.78 (297)	2.01 (858)
	XN20	9.99 (55)	0.78 (292)	0.98 (860)

WBC: White blood cells; RBC: Red blood cells; Hb: Haemoglobin; Hct: Hematocrit; PLT: Platelets; MVC: mean corpuscular volume; MCH: mean corpuscular hemoglobin. In the table CV% and mean for all parameters tested (Normal distribution of all CV% results included).

**Table 3 diagnostics-11-01756-t003:** Agreement between DxH 900 and XN20; slope and intercept results for degree of association for each parameter analyzed.

Parameters	N	Slope (a)	Slope 95% CI	Intercept (b)	Intercept 95% CI	Spearman Rho
WBC	979	0.949	[0. 945; 0. 954]	0.037	[0.011; 0.061]	0.996
RBC	987	0.939	[0.935; 0.943]	0.117	[0.098; 0.133]	0.997
Hb	987	0.964	[0.960; 0.967]	0.511	[0.461; 0.560]	0.998
Hct	987	0.977	[0.970; 0.986]	0.231	[−0.080; 0.532]	0.991
MCV	987	1.041	[1.023; 1.059]	−2.158	[−3.797; −0.517]	0.959
RDW	982	1.063	[1.040; 1.085]	−0.346	[−0.642; −0.008]	0.965
PLT	974	0.953	[0.946; 0.960]	1.463	[−0.055; 2.988]	0.991
MPV	929	1.003	[1.000; 1. 034]	−2.100	[−2.672; −2.100]	0.965

WBC: White blood cells; RBC: Red blood cells; Hb: Hemoglobin; Hct: Hematocrit; MCV: mean corpuscular volume, RDW: red cell distribution width; PLT: Platelets; MPV: mean platelet volume.

**Table 4 diagnostics-11-01756-t004:** Comparison of the DxH 900 results for the leukocyte differential to the reference method (optic microscopy); slope and intercept results for the degree of association for each parameter analyzed DxH 900 = a × Manual Diff + b.

Parameters	N	Slope (a)	Slope 95% CI	Intercept (b)	Intercept 95% CI	Spearman Rho
Neutrophils	976	0.946	[0.928; 0.961]	0.500	[−0.610; 1.580]	0.940
Lymphocytes	973	0.925	[0.907; 0.942]	3.248	[2.759; 3.747]	0.918
Monocytes	975	1.012	[0.964; 1.078]	1.625	[1.275; 1.988]	0.700
Eosinophils	972	0.985	[0.937; 1.000]	0.152	[0.100; 0.209]	0.834

**Table 5 diagnostics-11-01756-t005:** Comparison of the XN20 for the leukocyte differential to the reference method (optic microscopy); slope and intercept results for the degree of association for each parameter analyzed Sysmex XN20 = a × Manual Diff + b.

Parameters	N	Slope (a)	Slope 95% CI	Intercept (b)	Intercept 95% CI	Spearman Rho
Neutrophils	987	0.948	[0.932; 0.966]	0.271	[–0.792; 1.442]	0.943
Lymphocytes	985	0.919	[0.901; 0.937]	3.591	[3.10; 4.124]	0.927
Monocytes	987	0.980	[0.923; 1.032]	1.836	[1.480; 2,187]	0.693
Eosinophils	986	1,00	[0.941; 1,000]	0.200	[0.10; 0.200]	0.833

**Table 6 diagnostics-11-01756-t006:** Blast flag detection (reference method cut-off) sensitivity and specificity in the DxH 900 and the XN20.

Abnormal Cell	True Positive	False Positive	True Negative	False Negative	Sensitivity (%)	Specificity (%)	Balanced Accuracy (%)	Kappa
Blast manual count > 1% (n = 17)								
XN20	15	130	766	2	88.2	85.5	86.8	0.16
DxH 900	8	10	886	9	47.1	98.9	73.0	0.45
Blast manual count > 5% (n = 8)								
XN20	8	137	768	0	100	84.9	92.4	0.09
DxH 900	7	11	894	1	87.5	98.8	93.1	0.53
Blast manual count > 20% (n = 7)								
XN20	7	138	768	0	100	84.8	92.4	0.08
DxH 900	6	12	894	1	85.7	98.7	92.2	0.47

**Table 7 diagnostics-11-01756-t007:** Abnormal lymphocyte detection (reference method cut-off) and variant lymph: sensitivity and specificity in the DxH 900 and the X20N.

Abnormal Cell	True Positive	False Positive	True Negative	False Negative	Sensitivity (%)	Specificity (%)	Balanced Accuracy (%)	Kappa
Abnormal Lymph > 5% and Lymphocytosis > 3.5 × 10^9^/L								
XN20	18	127	763	7	72	85.7	78.9	0.17
DxH 900	16	46	844	9	64	94.8	79.4	0.34
Abnormal Lymph > 5% and Lymphocytosis > 5 × 10^9^/L								
XN20	18	127	763	7	72	85.7	78.9	0.17
DxH 900	14	51	878	9	85.7	94.5	90.1	0.29
Abnormal Lymph > 5% and Lymphocytosis > 50% (n = 15)								
XN20	14	131	768	2	87.5	85.4	86.5	0.15
DxH 900	15	47	852	1	93.8	94.8	94.3	0.37

**Table 8 diagnostics-11-01756-t008:** Reactive lymphocyte detection (reference method cut-off) and variant lymph: sensitivity and specificity in the DxH 900 and the XN20.

Abnormal Cell	True Positive	False Positive	True Negative	False Negative	Sensitivity (%)	Specificity (%)	Balanced Accuracy (%)	Kappa
Reactive Lymph > 5% and Lymphocytosis > 3.5 × 10^9^/L								
XN20	5	4	558	350	1.4	99.3	50.3	0.01
DxH 900	35	27	535	320	9.9	95.2	52.5	0.06
Reactive Lymph > 5% and Lymphocytosis > 5 × 10^9^/L								
XN20	5	4	582	326	1.5	99.3	50.4	0.01
DxH 900	35	27	559	296	10.6	95.4	53.0	0.07
Reactive Lymph > 5% and Lymphocytosis > 50%								
XN20	5	4	854	54	8.5	99.5	54.0	0.13
DxH 900	30	32	826	29	50.9	96.3	73.6	0.46

**Table 9 diagnostics-11-01756-t009:** Platelet clumps flag sensitivity and specificity in the DxH 900 and the XN20 combined with thrombocytopenia (<100,000/µL and <60,000/µL).

Abnormal Cell	True Positive	False Positive	True Negative	False Negative	Sensitivity (%)	Specificity (%)	Balanced Accuracy (%)	Kappa
Platelet clumps and Thrombocytopenia < 100.000/µL								
XN20	6	10	963	14	30.0	99.0	64.5	0.32
DxH 900	17	15	958	3	85	98.5	91.7	0.65
Platelet clumps and Thrombocytopenia < 60.000/µL								
XN20	4	12	978	5	44.4	98.8	71.6	0.31
DxH 900	9	23	967	0	100	97.7	98.8	0.43

## Data Availability

The datasets of this study are available from the corresponding author upon reasonable request. Requests to access the datasets should be directed to mserrando.girona.ics@gencat.cat.

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
