# Peer review of "Evaluation of the New Beckmann Coulter Analyzer DxH 900 Compared to Sysmex XN20: Analytical Performance and Flagging Efficiency"

_diagnostics, 2021, doi:10.3390/diagnostics11101756_

Round 1

Reviewer 1 Report

Thank you for performing the changes to the manuscript.

Author Response

Thank you very much for your comments and reviews. 

Reviewer 2 Report

I believe this is a big improvement but still many sentences are hard to understand and don't flow with the rest of the paragraph.  Some groups of small paragraphs discussing the same content would be best combined or elaborated with more detail.

19 For Platelets, XN20 demonstrated a negative bias in all samples analyzed.

20 DxH 900 and XN20 process samples differently but both are clinically accurate and efficient.

21 Abnormal Lymphocyte detection highlighted the differences between the two technologies as only minimal agreement was obtained.

42 platelet identification (PLTs)

72 two systems by determining CBC and WBCs differentiation

82 put %s in parenthesis: 12-20 years-old (8.5%); 21-40 years-old (14.3%);…

116  We  defined thrombocytopenia as platelet counts (PLT)  below <100,000/μL

121-125 Perhaps this could be explained more clearly.  Consider breaking up the sentences into several smaller sentences

141 Between day analysis

144 This sentence refers to values in Table 2 not 1: (CV=9.99%), showing higher differences than ought to be observed. No differences were 144

observed between values obtained from either analyzer (p>0.05) (Table 1).

Table 6 & 7 The false positive difference isn’t mentioned in the narrative

samples discussed in the narrative don’t always follow the order  presented in the referenced tables

a few sentences explaining the purpose of each statistical application would increase readability

It seems some detail in the M&M is lacking

Numerous typos spread throughout.  Check capitalizations too.

Some apparent differences in performance (such as False Positives) isn't mentioned in the Discussion.  

Author Response

RESPONSE TO REVIEWER 2 COMMENTS

Thank you very much for your suggestions and comments. We did the changes required in order to improve our article. 

  1. Abstract

We changed lines 19,20,21 and 42 according to your comments. The results text reviewed: 

Results: There was a strong correlation between the analyzers in almost all clinical parameters tested. Both DxH 900 and XN20 showed an excellent degree of association for the leukocyte differential compared to the reference method (manual microscopy). When it comes to IG%, XN20 showed a positive bias of higher results. Related to platelets, there are no differences between the two methods for PLT count. For mean platelet volume (MPV), DxH 900 provided 100% results of the samples analyzed while XN20 had a 16% result of that parameter left.  

  1. Material and Methods (parenthesis, thrombocytopenia definition and 121-125 clarification).

We did all the changes suggested. The text 2.2 flagging performance has been reviewed as:

2.2 Flagging performance

We performed the analysis of the DxH 900 and the XN20 focusing on their capability of flagging samples in the presence of abnormal cells. We included common analyzer flags such as blast, immature granulocytes (IG%) and platelet clumps. For the detection of abnormal lymphocytes each analyzer has specific flags. The variant lymph high sensitivity flag was used for the DxH 900 whereas blast/abn lymph and atypical lymph flags were used for the XN20. If the blast/abn Lymph is present, the XN20 performs subsequent analysis in the white precursor cell channel (WPC) that allows the analyzer discrimination between blast and abnormal cells detection. In this study these two XN20 flags were assessed together to ensure homogeneity with DxH 900 that only has one flag for the abnormal lymphocyte detection. We did not included flags from de WPC in the XN20 performance.

  1. Mixed results in table 1 and 2

Changed and reviewed. 

  1. False positive results in table 6 and 7; they were not mentioned in the narrative

We included these results in the narrative and in the discussion according to the reviewer suggestions. 

  1. Numerous typos, capitalizations

We checked on all of these mistakes and tried to correct all of them.

  1. Few sentences explaining the purpose of each statistical application

This information has been added in the M-M narrative: "We used R studio open software for statistics. Passing-Bablok regression and Bland-Altman analysis were performed to compare both analyzers (results from the two analyzers are plotted against the median of the two techniques). Spearman’s test was used to obtain the correlation coefficient. For the basophile count we performed a Chi test".

We would like to thank you for your suggestions and comments; it has been really helpful in order to improve our skills as scientists. 

Reviewer 3 Report

The paper of Querol and colleagues compared new Beckmann Coulter Analyzer DxH 900 and Sysmex XN20 in terms of their analytical performance and flagging efficieny. Authors proved a strong correlation between the analyzers in almost all clinical parameters tested. However, authors revealed also some differences, showning a positive bias of higher results of Ig% as well as a negative bias of platelets when Sysmex XV20 was used. This paper may constitute a significant clinical value when applyong one of these analyzers in diagnostic laboratory tests.

Regardless of the scientific significance of the publication's results I have raised some comments: 

Comment 1: Authors claimed in the abstract that: "DxH 900 and XN20 were significantly different but clinically accurate and efficient". I am not sure if we can say that both analyzers are significantly different, rather authors should use such terms when describing the results obtained from both apparatus. 
Also term: "clinically accurate and efficient" should be avoided sonce both DxH 900 and XN20 are diagnostic equipments and their clinically accurate and efficient were walidated and optimised earlier but not in the present study.

Comment 2: There are many fragments of text highlighted in yellow but there is no explanation why - are these some fragments after language correction or after revision in other journal? Authors should clarify this.

Round 2

Reviewer 3 Report

In my opinion authors made corrections according to my and other Reviewers' sugestions. However i noticed that the quality of figures presenting obtained results is very poor like these are raw data obtained directly from analyzers but these figures do not look professional statistical graphs. It should be corrected.

Author Response

Thank you very much for your suggestion. We corrected the figures attached in the text in order to improve their quality and scientific content. 

This manuscript is a resubmission of an earlier submission. The following is a list of the peer review reports and author responses from that submission.

Round 1

Reviewer 1 Report

extensive grammar & formatting errors.  No references are provided.  

No keywords listed

56 "which is used"

108 many "?" 

120 rephrase to avoid beginning with 95

149 clarify 

154 fix page break

170-174 strange format, capitalizations and italics

179 rephrase

200 incomlete

210 rephrase "are not good"

254 replace comma w/ a period

273 "compared to" check parenthesis

287 rephrase

324 "the difference between the 2 analyzers was expected since there is no MPV reference method for standardization"

334 replace comma w/ a period

Reviewer 2 Report

The authors have submitted a meaningful manuscript describing the efficiency and accuracy in two automated hematology analyzers. This information is potentially useful to clinical pathologists and laboratory technicians all around the world, and I believe it could be worthy of publication. Nonetheless, the manuscript presents several grammar mistakes, is hard to read and has major problems in its structure (The “Results” section should only state objective data and any explanation or discussion should be moved to “Discussion”). Moreover, I believe there are many data that could be interesting and are not provided (CVs for many parameters) and it seems that some Tables are incomplete.

I have a few questions, comments, and suggestions for the authors, which I've listed below, by line number in the manuscript:

Introduction

Please, double-check the entire manuscript and correct some verbs, plurals, grammar incongruencies and mistakes (for example, see doubled sentence in Line 55). There are large sections of the manuscript that require English edition.

Line 55.- This sentence “Cells are analysed in near-native state, in an environment as close to the human body as possible and without stain” needs further explanation or clarification.

Materials and Methods:

Line 82.- How many pediatric samples were present?. Were newborns included?. Since pediatric samples can markedly differ from adult ones, it would be interesting to sub-divide the group 0-20 years and re-analyze in other to discard any age effect.

Line 87  It would be interesting to detail the specific software version in each analyzer (or at least to clarify that they were updated following the manufacturers` instructions).

Line 95.- Did the authors use an automatic stainer or were the samples manually stained?.

Line 96.- It would be better to acknowledge which authors performed the manual examination. Was this exam performed by the authors, by technicians or by clinical pathologists?. Were the same authors for every sample?.

Line 116.- It is a pity not to present all your results and only be focused on WBC (in general), RBC and platelets for the analytical performance tests. It would be markedly more informative to show the results for each leukocyte subpopulation and erythrocyte index, since more information could be of interest for reference laboratories.

Line 121.- This sentence is difficult to understand.

Line 125.- Please state which Statistics software was used for this study.

Results:

Table 1 – Were data normally distributed? (since the authors report means).

Line 149.- This sentence is redundant and difficult to understand.

Line 153.- “Average” is not a correct term (either mean or median, depending on the distribution of your data).

Figure 1.- Do not express personal beliefs or conclusions in the legend; legends should be only descriptive.

Line 159.- It is strange to reference here MCV, MCH or hemoglobin when there are not references to these parameters previously.

Line 160.- This section should only provide data and objective findings. Any belief or explanation for your results should be moved to Discussion.

Line 163.- Font in this sentence is different (there are also various paragraphs mistakes in the manuscript). Please be consistent using your tenses (either past or present).

Table 3.- It is unclear what this table is displaying (which data are here compared to optical differentials? Those obtained using Sysmex XN? Dxh900? Both? How is it possible that both analyzers showed similar slopes?. I think the Table in incomplete, since the last sentence of the legend is strange.

Line 188.- Again, Results should only provide objective data and information, not explanations.

Line 203.- This sentence is more appropriate for Discussion, not for Results.

Figure 3.- Similar comments to Figure 1.

Discussion:

Line 342.- Please discuss whether including flagged samples in your analytical studies could have influenced your results.